# The Anti-Inflammatory Effect of Nanoarchaeosomes on Human Endothelial Cells

**DOI:** 10.3390/pharmaceutics14040736

**Published:** 2022-03-29

**Authors:** Nancy Charó, Horacio Jerez, Silvio Tatti, Eder Lilia Romero, Mirta Schattner

**Affiliations:** 1Laboratory of Experimental Thrombosis and Immunobiology of Inflammation, Institute of Experimental Medicine, CONICET-National Academy of Medicine, Pacheco de Melo 3081, Buenos Aires 1425, Argentina; nancycharo@hotmail.com; 2Center for Research and Development in Nanomedicines (CIDEN), National University of Quilmes, Roque Saenz Peña, Bernal 1876, Argentina; jairjerez@gmail.com; 3Department of Obstetrics and Gynecology, Clinical Hospital, Av. Córdoba 2351, Buenos Aires 1120, Argentina; drestatti@gmail.com

**Keywords:** nanoarchaeosomes, endothelium, inflammation, ICAM-1, von Willebrand factor, nanovesicles

## Abstract

Archaebacterias are considered a unique source of novel biomaterials of interest for nanomedicine. In this perspective, the effects of nanoarchaeosomes (ARC), which are nanovesicles prepared from polar lipids extracted from the extreme halophilic *Halorubrum tebenquinchense*, on human umbilical vein endothelial cells (HUVEC) were investigated in physiological and under inflammatory static conditions. Upon incubation, ARC (170 nm mean size, −41 mV ζ) did not affect viability, cell proliferation, and expression of intercellular adhesion molecule-1 (ICAM-1) and E-selectin under basal conditions, but reduced expression of both molecules and secretion of IL-6 induced by lypopolysaccharide (LPS), Pam3CSK4 or *Escherichia coli*. Such effects were not observed with TNF-α or IL-1β stimulation. Interestingly, ARC significantly decreased basal levels of von Willebrand factor (vWF) and levels induced by all stimuli. None of these parameters was altered by liposomes of hydrogenated phosphatidylcholine and cholesterol of comparable size and concentration. Only ARC were endocytosed by HUVEC and reduced mRNA expression of ICAM-1 and vWF via NF-ĸB and ERK1/2 in LPS-stimulated cells. This is the first report of the anti-inflammatory effect of ARC on endothelial cells and our data suggest that its future use in vascular disease may hopefully be of particular interest.

## 1. Introduction

In the last 25 years, nearly 60 therapeutic nanomedicines entered the pharmaceutical market, while a growing number are in advanced clinical trials [1], most of them aimed to target solid tumors upon intravenous administration [2]. The recent approval by the FDA of Patisiran/Onpattro, a novel nanomedicine for efficient RNA delivery to the liver [3], has paved the way for the fast advent of new nanoparticulate anti-COVID-19 vaccines, such as those from Moderna and Pfizer [4]. However, despite cardiovascular disease being the primary cause of global mortality (31% of all deaths worldwide) [5], the evolution of cardiovascular nanomedicines is still in its childhood [6]. Research in this area consists of proof-of-concept nanomedicines targeted to the vascular endothelium and neighbouring cells to deliver anti-inflammatory or anti-proliferative agents [7]. The endothelium is a key target for therapeutic interventions in several conditions. It represents an enormously extended surface area accessible to blood (3000–6000 m^2^) [8], and is not only critical for the maintenance of vascular function and homeostasis, but it is also sensitive to the chemical nature of nanoparticles (NPs) in the bloodstream. In fact, endothelial injury or dysfunction is the initial cause of atherosclerosis and other cardiovascular diseases [9]. For instance, endothelial cells are sensitive to the chemical nature of metallic and metallic oxide NPs. ZnO, silica, and Ag NPs activate human endothelial cells through the NF-κB pathway [10,11,12,13]. Gold NPs, however, are less harmful to endothelial cells, and have been employed in the identification of plaques and the recognition of inflammatory markers [14].

The endothelium, on the other hand, seems to be inert to the interaction with biodegradable polymer NPs, micelles, and liposomes conforming to the core of most therapeutic and prophylactic nanomedicines [15,16]. New materials of a heterogeneous chemical nature, such as inorganic-organic hybrids, whether bio-inspired [17], or of natural origin, are increasingly enriching the architecture of novel preclinical nanomedicines, but whose effect on the vascular endothelium are still not completely clear. Nanoarchaeosomes (ARC), for instance, are nanovesicles in current preclinical exploration as drug and vaccine nanocarriers, consisting of closed bilayers made of amphipathic archaeolipids isolated from extreme halophiles, methanogens or extreme thermophile archaebacteria [18,19,20,21,22,23]. The nature of the hydrophilic headgroups and the hydrophobic fully saturated chains of polar archaeolipids depend on the archaebacteria genus [24], meaning that the term ARC encompasses nanovesicles of variable structural composition. Different from phospholipids from Eukarya and Bacteria domains, the amphipathic archaeolipid possesses an sn-2,3 stereoisomerism, and a glycerol backbone ether-linked to fully saturated polyisoprenoid chains. Such unique chemical structure makes ARC refractory to enzymatic attacks, hydrolysis and oxidation [25], cold chain interruption during transport and storage [26], and resistant to nebulization shear stress [27]. Archaebacteria’s lack of lipopolysaccharide (LPS) and ARC from different sources showed to be non-toxic or mildly toxic in in vitro and in vivo conditions [28]. Because of these reasons, ARC are interesting alternatives to liposomes, are less labile, and have extended structural benefits. For example, we have recently reported that ARC from *Halorubrum*
*tebenquichense* (*H. tebenquichense*) loaded with dexamethasone or curcumin display repairing activity on inflamed intestinal and lung epithelia [29,30]. The specific composition of *H. tebenquichense* ARC was determined by electrospray-ionization mass spectrometry (ESI-MS) [31] and ordered according to decrescent abundance as archaeol analog methyl ester of phosphatidylglycerol phosphate (PGP-Me), archaeol analog phosphatidylglycerol (PG), (1-O-[a-D-mannose-(2′-SO3H)-(1′ a 2′)-a-D-glucose]-2,3-di-O-phytanyl-sn-glycerol) (SDGD5) the cardiolipin bis phosphatidylglycerol (BPG) and the glycocardiolipin SDGD-5PA (2′-SO3H)-Manp-a1,2Glcpa-1-1-[sn-2,3-di-O-phytanylglycerol]-6-[phospho-sn-2,3-di-O-phytanylglycerol]. Until now however, there are no reports about the interaction of these ARC with endothelial cells. We present here the first assessment of ARC uptake by human endothelial cells under physiological and inflammatory conditions.

## 2. Materials and Methods

Hydrogenated soy phosphatidylcholine (HSPC) was purchased from Northern Lipids Inc (Vancouver, BC, Canada). Cholesterol was from ICN Biomedicals. Lissamine™ rhodamine B 1,2-dihexadecanoyl-sn-glycero-3-phosphoethanolamine triethylammonium salt (RhPE) was obtained from Thermo Fisher Scientific (Waltham, MA, USA). Yeast extract was from Laboratorios Britania S.A. (Buenos Aires, Argentina). Purified LPS derived from *Escherichia coli* (*E. coli*) O111:B4, phorbol 12-myristate 13-acetate (PMA), and Ionomycin were purchased from Sigma (Burlington, MA, USA). Pam3CysSerLys4 (Pam3CSK4) and polyinosinic: polycytidylic acid (Poly (I: C)) were from InvivoGen (San Diego, CA, USA). Recombinant Human Tumor Necrosis Factor-alpha (TNF-α) was purchased from Cell Signaling Technology (Beverly, MA, USA). Interleukin (IL)-1β and Click-iT^®^ Plus EdU Alexa Fluor-488, were obtained from Invitrogen (Carlsbad, CA, USA). *E. coli* DH5a was a kind gift from Dr. Ricardo Gomez (Biotechnology and Molecular Biology Institute, CONICET-UNLP, La Plata, Argentina). The other analytic grade reagents were from Anedra, Research AG (Buenos Aires, Argentina).

### 2.1. Archaebacteria Growth, Extraction & Characterization of Total Polar Archaeolipids

*H. tebenquichense*, archaea isolated from soil samples collected from Salina Chica, Peninsula de Valdés, Chubut, Argentina were grown in basal medium supplemented with yeast extract and glucose at 37 °C [19]. Biomass was grown in a 15 L medium in a 25 L homemade stainless-steel bioreactor and harvested after 96 h growth. Total polar archaeolipids (TPA) were extracted from the biomass using the method of Bligh and Dyer modified for extreme halophiles [32]. Between 400 mg and 700 mg of TPA was isolated from each culture batch. The reproducibility of each TPA-extract composition was routinely screened by phosphate content [33] and ESI-MS, as described in [31] (Appendix A).

### 2.2. Preparation of Nanovesicles

Conventional liposomes made of HSPC: cholesterol 7.5:2.5 *w*/*w* (LIP), and nanoarchaeosomes made of TPA: cholesterol (7:3: *w*/*w*) of the extreme halophile archaea *H. tebenquichense* (ARC), were prepared by the film hydration method [34]. Total lipids were dissolved in chloroform: methanol 1:1 *v*/*v* and solvents were rotary evaporated at 40 °C until elimination. The lipid films were rinsed with N_2_ and hydrated with 10 mM Tris-HCl buffer pH 7.4, 0.9% *w*:*w* NaCl up to a final concentration of 10 mg/mL total lipids. The suspensions were sonicated for 1 h with a bath-type sonicator 80 W, 80 KHz and extruded 10–15 times through polycarbonate filters with a pore size of 0.4 μm and 0.2 μm using a Thermobarrel extruder (Northern Lipids, Vancouver, BC, Canada). For the preparation of LIP, hydration, sonication, and extrusion were performed at 55–60 °C. To prepare RhPE-labeled nanovesicles, RhPE was added to the mixed organic solution of lipids at a rate of 0.4 μg per mg of lipids and the resultant lipid films were hydrated with a NaCl-Tris-HCl buffer as described above. All nanovesicles were sterilized by passage through a 0.22 μm sterile filter and stored at 4 °C.

### 2.3. Characterization of Nanovesicles

Total phospholipid content was quantified by colorimetric phosphate microassay [33]. RhPE was quantified by spectrofluorometry (λ_ex_ = 561 nm and λ_em_ = 580 nm) using a LS55 fluorescence spectrometer (PerkinElmer Inc., Waltham, MA, USA), after complete disruption of 1 volume of nanovesicles in 10 volume of methanol. The fluorescence intensity of the sample was compared with a standard curve prepared using RhPE in methanol. The standard curve was linear in the range of 0.075–0.5 μg/mL RhPE with a correlation coefficient of 0.999. The size and ζ potential of nanovesicles were determined by dynamic light scattering and phase analysis light scattering, respectively, using a Zeta nanosizer instrument (Malvern Instruments, Malvern, Worcestershire, UK).

### 2.4. Endothelial Cell Culture

This study conforms to the tenets of the Declaration of Helsinki and was previously approved by the institution’s ethics committee. The umbilical cord was collected after normal deliveries with written informed consent from the mother. Human umbilical vein endothelial cells (HUVEC) were purified from human umbilical vein by digestion with collagenase (Gibco, Grand Island, NY, USA) according to the method of Jaffe et al. [35]. Cells were grown in a 2% gelatin (Sigma, St. Louis, MO, USA) coated plate with Endothelial Growth Medium-2 (EGM2) (Lonza, Lexington, MD, USA) supplemented with antibiotics (100 U/mL penicillin and 100 µg/mL streptomycin). HUVEC were used between the first and fourth passages. Human microvascular endothelial cells (HMEC) (ATCC Cell Lines) were maintained in MCDB 131 medium containing 15% *v*/*v* fetal bovine serum, hydrocortisone, and human epidermal growth factor human supplemented with antibiotics. Cells were incubated at 37 °C in a humidified atmosphere with 5% CO_2_.

### 2.5. Measurement of Cell Viability and Apoptosis

HUVEC cultured in RPMI/1% FBS (Gibco) were treated in the absence or presence of increasing concentrations of nanovesicles, and cell viability was determined 24 h later. Adherent cells were harvested with 0.25% trypsin and pooled with detached cells. Cells were then stained with a mixture of ethidium bromide and acridine orange (100 µg/mL), mounted on slides, and immediately analyzed by fluorescence microscopy. At least 300 cells per treatment were counted [36].

### 2.6. EdU Cell Proliferation Assay

HUVEC (2.5 × 10^4^ cells) were plated in 48-well plates and incubated overnight at 37 °C to allow attachment. Cells were then incubated with ARC in EGM2 for 24 and 48 h. Subsequently, cells were incubated with 10 µM 5-ethynyl-2′-deoxyuridine (EdU) for 2 h. After incubation, cells were trypsinized, washed with phosphate buffered saline (PBS), and then stained with Alexa fluor 780 viability dye (eBioscience) at 4 °C for 30 min. Cells were fixed with 4% paraformaldehyde (PFA) for 15 min, washed once with bovine serum albumin/PBS buffer, permeabilized with a perm/wash buffer (BD Biosciences, San Jose, CA, USA), and treated with Alexa Fluor^®^ 488 using the Click-iT^®^ EdU Imaging Kit, washed, resuspended in buffer and analyzed by flow cytometry using a Sysmex-Partec CyFlow^®^ Space. The percentage of proliferating cells that have incorporated EdU was defined within the viable cell population.

### 2.7. Expression of E-Selectin and ICAM-1

HUVEC were incubated with or without ARC and various stimuli. ICAM-1 expression was determined after 18 h by labelling cells in the dark with phycoerythrin (PE)-mouse anti-human CD54 (clone HA58, BD Pharmingen, San Diego, CA, USA). Because of the different expression peak ([37,38]), E-selectin was evaluated after 4 h HUVEC stimulation and then stained with a PE-anti-CD62E monoclonal antibody (Ab) (Clone 68-5H11, BD Pharmingen). To determine the nonspecific binding, cells were labeled with irrelevant isotype-matched IgG1 (Clone MOPC-21, BD Pharmingen). After labelling, cells were washed, fixed with 1% PFA, and then analyzed by flow cytometry.

### 2.8. Measurement of IL-6 and vWF

IL-6 was measured in the supernatant of the cells using an ELISA kit (eBioscience, San Diego, CA, USA). The concentrations of von Willebrand factor (vWF) in the supernatants were determined with a home-made ELISA using human vWF and horseradish peroxidase (HRP)-conjugated human vWF as primary and secondary Abs, respectively (Dako, Glostrup, Denmark). Results were expressed in ng/mL and extrapolated from serial dilutions of pooled normal plasma, assuming a vWF concentration of 7 μg/mL [39].

### 2.9. ARC Uptake by HUVEC

HUVEC (5 × 10^4^ cells/well) were incubated with 50 μg/mL RhPE-labeled nanovesicles for various periods at 37 and 4 °C (control for nonspecific adsorption on the cell surface) and then trypsinized, washed, and analyzed by flow cytometry. The fluorescence of cells treated with ARC were corrected using the autofluorescence of untreated cells and the resulting data were analyzed using FlowJo 7.6 software. For confocal microscopy experiments, HUVEC were seeded on a glass coverslip and incubated with RhPE-labeled ARC for 4 h. Cells were then washed three times with cold PBS, fixed with 4% PFA, and stained with fluorescein IsoTioCyanate-phalloidin. After 3 washes, slides were inverted, mounted with PolyMount and analyzed by fluorescence microscopy using a FV-1000 microscope (Olympus, Tokyo, Japan).

### 2.10. Analysis of ERK1/2 and NF-κB Signaling Pathways

HMEC (2 × 10^5^ cells/well) were cultured in complete medium and incubated overnight at 37 °C to allow attachment. HMEC were exposed to serum starvation for 2 h and then stimulated with LPS in the absence or presence of ARC. After incubation, endothelial cells were trypsinized, washed, fixed, and permeabilized as described in the BD Phosflow protocol (protocol III). Subsequently, cells were labeled with Abs against phospho-ERK1/2 (Tyr 202/Tyr 204) or IgG1κ isotype conjugated with Alexa Fluor^®^ 488 and subjected to flow cytometry. NF-κB signaling was assessed by Western blotting. After starvation, HMEC were scraped and lysed with lysis buffer (60 mM de Tris/HCl, pH 6.8 + 1% SDS) in the presence of a cocktail of protease inhibitors (Sigma). After centrifugation, the lysates were electrophoresed and transferred to a nitrocellulose membrane. After blocking membranes were incubated with anti-phospho-ERK1/2 Santa Cruz Biotechnology, Santa Cruz, CA, USA) and anti-IκBα (BD Biosciences) followed by HRP-conjugated secondary Ab. Each membrane was reprobed with an Ab against β-actin (BD Biosciences) and proteins bands were visualized by ECL.

### 2.11. Quantitative PCR (qPCR)

HMEC were stimulated with LPS (1 μg/mL) with or without ARC for 18 h at 37 °C. Then, cells were washed and harvested with Trizol (Life Technologies, Carlsbad, CA, USA). Reverse transcription was performed using 100 ng of RNA and the iScript cDNA synthesis kit (Bio-Rad, Hercules, CA, USA). Real-time PCR reactions were determined using 1 μL cDNA and the SsoAdvanced universal SYBR Green mix and CFX-Connect equipment (Bio-Rad). The primers used are described in the Appendix A. The reaction was normalized to the expression levels of the housekeeping gene, and the specificity of the amplified products was verified by dissociation curves analysis.

### 2.12. Statistical Analysis

Results were reported as means ± SEM of independent experiments. The Shapiro-Wilk test was used to define the condition of normality and equal variance. Statistical differences between means were determined with Student’s paired *t*-test or One-way ANOVA followed by Bonferroni multiple comparison tests using Graph Pad Prism version 6.00 (GraphPad Software Inc., San Diego, CA, USA). *p*  <  0.05 values were considered statistically significant.

## 3. Results

### 3.1. Structural Characterization

The structural features (size, polydispersity index, ζ potential, total lipids, and RhPE) are shown in Table 1. Table 1 shows the structural features of the resulting nanovesicles, at concentrations between 7.5 ± 1.3 and 6.9 ± 0.6 for nanoARC and nanoARC-RhPE and 7.0 ± 1.7 and 6.0 ± 0.3 mg lipids/mL for LIP and LIP-RhPE, respectively. The nanoarchaeosomes exhibited a strong negative ζ potential, with mean diameters in the order of 160 nm, almost 100 nm smaller than those of liposomes. Both the nanoarchaeosomes and liposomes exhibited low PDI (≤0.3) and were labelled at a comparable ratio of RhPE/total lipids between 0.48–0.45 μg Rh-PE/mg TL.

### 3.2. ARC Do Not Alter HUVEC’s Viability or Proliferation

Initially, we analyzed the effect of ARC on HUVEC viability and apoptosis by staining cells with ethidium bromide and acridine orange. As shown in Figure 1A, neither ARC nor liposomes affect the viability and apoptosis of HUVEC after 24 h incubation at concentrations up to 50 μg/mL. However, a statistically significant reduction in the cell viability and nuclear changes characteristic of apoptotic cells were observed in HUVEC treated with 100 μg/mL ARC (Figure 1A,B). Thus, in all the rest of the experiments, ARC and liposomes were used at 50 μg/mL. To determine whether ARC alter HUVEC growth, the cell cycle was evaluated by measuring the EdU incorporation. No differences in the thymidine analog incorporation were observed regardless of whether cells were exposed to nanovesicles or not (Figure 1C).

### 3.3. ARC Selectively Inhibit the Expression of E-Selectin and ICAM-1

We next investigated the effect of ARC on different endothelial cell inflammatory and hemostatic responses. Figure 2A shows that basal expression levels of the cell adhesion molecules, E-selectin and ICAM-1, were not modified by the presence of ARC. However, exposure of endothelial cells to ARC reduced the levels of E-selectin or ICAM-1 expression triggered by LPS, Pam3CSK4, or *E coli* (Figure 2B–D). ARC inhibited ICAM-1 expression in *E.coli* stimulated cells with an IC50 = 9.9 ± 1.91 μg/mL. Intriguingly, E-selectin and ICAM-1 expression levels in TNF-α or IL-1β stimulated cells were not modified by the ARC (Figure 2E,F). On the other hand, the expression of both endothelial cell adhesion molecules induced by any stimuli (Figure 2B–F) was similar regardless of their exposure or not to liposomes. A decreased expression of ICAM-1 mediated by ARC was also observed when HMEC were used instead of HUVEC (Appendix A). Moreover, qPCR studies showed that the inhibitory effect was associated with a significant reduction in the mRNA levels (Appendix A).

### 3.4. ARC Decrease the Levels of Cytokines Released by Endothelial Cells Stimulated with LPS or Pam3CSK4 but Not with TNF-α or IL-1β

Having demonstrated an inhibitory effect of ARC on endothelial cell adhesion molecules expression, we next evaluated the effect of ARC on the production of pro-inflammatory IL-6. As it is shown in Figure 3A,B, IL-6 released by LPS or Pam3CSK4 was significantly decreased by the presence of ARC but not by liposomes. However, and similarly to the results obtained in the expression of cell adhesion molecules, ARC did not modify IL-6 secretion triggered by TNF-α or IL-1β (Figure 3C,D).

### 3.5. The Inhibitory Effect of ARC Is Not Due to Their Interaction with LPS, Pam3CSK4, or E. coli or Their Receptors

The observation that ARC inhibited endothelial cell activation by LPS, Pam3CSK4, or *E. coli* but not by TNF-α or IL-1β raised the question of whether there was an interaction between these stimuli and/or their receptors and whether ARC could lead to the inhibition of the agonist’s activity. Hence, HUVEC were pre-incubated with ARC for 1 h, then the medium was removed, the cells were washed and then stimulated for 18 h. Figure 4A–D shows that even under these experimental conditions, ICAM-1 expression triggered by LPS, Pam3CSK4, or *E. coli* but not TNF-α stimulated cells was reduced by ARC and not by liposomes. Moreover, we also observed that the augmentation of ICAM-1 or IL-6 levels triggered by activation of HUVEC with Poly (I: C), an agonist of endosomal TLR3, or by the combination of PMA plus Ionomycin, receptor-independent agonists, was also blocked by the presence of ARC (Appendix A).

### 3.6. ARC Decrease the Levels of vWF Released by Stimulated HUVEC

To determine whether ARC were also capable of reducing the prothrombotic activity of endothelial cells, we next examined its effect on two secretory pathways of vWF. vWF can be constitutively released during synthesis, or secreted from Weibel-Palade bodies (WPBs) in a regulated manner in response to various stimuli [40]. We found that the basal levels (Figure 5A) and the increased amount of vWF-triggered by LPS or Pam3CSK4 were markedly reduced by ARC (Figure 5B,C). Notably, similar results were obtained when cells were stimulated with TNF-α or IL-1β (Figure 5D,E). The reduction in vWF levels was also observed at the mRNA expression (Appendix A). Neither the release nor the synthesis of vWF was modified by the liposomes (Figure 5A–E).

### 3.7. ARC Are Internalized by HUVEC

Considering that the effect of nanovesicles is generally determined in part by how they are processed by cells, we decided to investigate the fate of ARC during their interaction with the endothelial cells. Hence, we used RhPE-labeled ARC for visualization and quantitative analysis of cellular uptake of the nanovesicles. The extent of uptake by cells was significantly higher for ARC than for liposomes at different times, and the maximum uptake efficiency occurred at 18 h (Figure 6A). By using confocal microscopy, we observed a high amount of RhPE-labeled ARC inside the cells while liposomes were poorly internalized (Figure 6B). The extent of the uptake was not modified in the presence of a stimulus (Appendix A).

### 3.8. ARC Attenuate LPS-Induced Activation of NF-κB Pathway

MAPK and NF-κB signaling are involved in the inflammatory response of HUVEC, including the expression of chemokines, cytokines, and adhesion molecules [41]. To understand the signaling pathway involved in the anti-inflammatory activity of ARC we analyzed the activation of the NF-κB pathway through IκBα degradation and phosphorylation of ERK1/2. Flow cytometry and Western blotting studies showed that phosphorylation of ERK1/2 induced by LPS was attenuated in the presence of ARC (Figure 7A,B). The IκBα degradation by LPS was also inhibited by ARC (Figure 7C) indicating that ARC modulate the inflammatory response in LPS-stimulated HMEC through NF-κB and the ERK1/2 signaling pathway. Neither phosphorylation of ERK nor IκBα degradation mediated by LPS were modified by liposomes (Figure 7A–C).

## 4. Discussion

A detailed comprehension of the processes and mechanisms underlying cellular NPs uptake is critical for exploring the effects of nanomaterials on biological systems. Moreover, evaluation of their potential harm to organisms is mandatory to promote a safer and more efficient application of nanomaterials in biomedical fields. In this study, we have analyzed the effect of ARC from *H. tebenquichense* on the endothelium and compared it with conventional liposomes. We found that these ARC, up to 50 μg/mL, exert a very strong anti-inflammatory effect on endothelial cells without altering cell viability or proliferation. Exposure of HUVEC to ARC markedly inhibited the expression and synthesis of cell adhesion molecules (CAMs) such as E-selectin and ICAM-1, the release of IL-6, and vWF. None of these effects were observed when endothelial cells were treated with conventional liposomes constituted by hydrogenated soybean phospholipids. Most of the in-vitro studies show that biodegradable nanocarriers do not have a direct effect on the endothelium [42,43,44] or that NPs exert an opposite effect compared to ARC on the endothelium, which means that they are pro-inflammatory [45,46]. So far, there is no evidence showing that NPs per se can inhibit endothelial cell activation responses.

The inhibitory effect of ARC in all effector-endothelial responses evaluated, except on the release of vWF, was observed when cells were stimulated with LPS, *E. coli*, or Pam3CSK4 (the two former ligands of Toll-like receptor (TLR) 4 and the latter of TLR2) but not when they were activated by cytokines such as IL-1β or TNF-α.

Taking into account that the signaling in response to all stimuli investigated in the current study has been shown to share the same molecular signals downstream from their receptors [47,48,49,50], it could be speculated that the inhibitory effects of ARC were specifically acting at the level of the agonist receptor complex or first target signaling proteins. Interference with efficient TLR agonist signaling by ARC at the level of the ligand-receptor could occur by direct interaction with TLR or accessory molecules, and interference with the assembly of the TLR receptor complex. In this regard, it has been shown that oxidized phospholipids antagonize the pro-inflammatory response to LPS but not by TNF-α and IL-1β, and its protective effect was ascribed to the inhibition of LPS binding to the LPS binding protein and CD14, resulting in the suppression of TLR-dependent signaling pathways [51]. Similarly, high-density lipoprotein interferes with LPS activity caused by the competitive binding of the LPS–LBP–CD14 complex to apoAI [52,53,54]. However, in our experiments, when ARC were incubated for 1 h and then removed before stimulation of the cells with LPS, Pam3CSK4, or *E. coli*, the anti-inflammatory effect persisted. Moreover, the inhibitory effect of ARC was also observed when endothelial cells were stimulated with PMA and Ionomycin (two agonists acting independently of receptor interaction) or with Poly (I: C) a ligand of the endosomal TLR3. Altogether these data suggest that a direct interaction between ARC and specific ligands of TLRs, their receptors, or accessory molecules is unlikely.

Several recent reports indicate that lipid microdomains favor the recruitment and clustering of the TLR machinery. Upon activation, TLRs migrate to specific lipid microdomains together with co-receptors where they are recruited, activated, and able to transmit signals [55,56,57]. In this regard, the down-regulating effect of surfactants or oxidized phospholipids on LPS activation was correlated to a diminished TLR4 translocation into lipid rafts [58,59]. Different from liposome bilayers, ARC bilayers are highly disorganized but with low lateral mobility and high local microviscosity [60,61,62]. Two remarkable features of archaeolipids are their structural resemblance with ramified iso-stearic acid and the exhibition of solubilizing and fluidity properties of oleic acid [63], which allow ARC bilayers to partition poorly soluble drugs [23,29]. Recently, the ability of archaeolipids to establish strong non-covalent bonds with cholesterol derivatives has been described [64]. Since we demonstrated that ARC are endocytosed, it could be conceivable that upon endocytosis, ARC downregulate endothelial activation responses induced by TLRs agonists by partitioning into the cell bilayer, sequestering cholesterol, and disrupting lipid rafts. This may be a relevant mechanism by which several lipid structures can modulate cellular responses.

PMA, on the other hand, is a phorbol ester that does not require receptors to reversibly activate Protein kinase (PK) C, a pathway ending in NF-κB activation. The PCK activation depends on its partition on the surface of PMA containing bilayers [65]. Therefore, a possible mechanism for PMA plus Ionomycin inhibition could be that archaeolipids in the cell membrane associate with the phorbol ester-binding domain of PKC, inhibiting its activation by PMA.

Finally, despite IL-1β and TNF-α signalling routes converging in NF-kB activation, the early events at the level of cell membrane differ from those that take place for TLRs. Upon ligand binding, IL-1β migrate into caveolae (flask-shaped invaginated microdomains rich in caveolin) instead of lipid rafts [66,67], whereas TNF-α mediated NF-kB activation, is reported to be independent of caveolae or lipid rafts [68]. Hence, the absence of anti-inflammatory effect upon IL-1β stimulation could be explained by a refractory character of caveolae to archaeolipid induced disorganization. On the other hand, because of its independence with early membrane events, the TNF-α stimuli would not be affected by archaeolipids.

Future experiments are necessary to elucidate the molecular basis of the agonist-selective-inhibitory effect of ARC on endothelial activation responses.

Induction of CAMs as well as the release of pro-inflammatory cytokines are controlled at the level of gene transcription and requires among the different molecular signals the binding of the transcription factor NF-kB to the regulatory region within the promoters of each of these genes [69]. Our data show that the inhibition of both NF-kB pathway activation and ERK phosphorylation are involved in the anti-inflammatory effect mediated by ARC. The role of NF-kB appears to be of major relevance not only to these ARC but also to the different NPs, such as those that are pro-inflammatory, that exert this effect through activation of this pathway.

Besides reducing the expression of CAMs and the release of cytokines, ARC markedly impaired the release and synthesis of vWF. The inhibitory effect was observed both in the acute and the constitutive release of vWF induced by all agonists tested. Since WPBs degranulation involves the rearrangement of cytoskeletal actin and myosin microfilaments, it could be possible that ARC interfere not only with lipid rafts assembly but also with cytoskeletal rearrangement. In this regard, it was reported that treatment of HUVEC with the anti-inflammatory lipid molecules such as eicosapentaenoic or docosahexaenoic acid impairs the release of vWF by attenuating acting reorganization [70]. WPBs are endothelial storage granules of vWF and other vasoactive molecules such as P-selectin and endothelin-1. Because the excessive secretion of these molecules contributes to inflammation related to hypertension and thrombosis, the blockade of WPBs exocytosis is critical to lessen endothelial dysfunction. The central role of vWF in thrombosis has made it a promising target for research into new antiplatelet therapies that inhibit vWF. It has been suggested that directly limiting vWF release from WPBs has the potential as a therapeutic for cardiovascular disease [71]. In addition, the expression of CAMs, as well as the release of IL-6 or IL-1β from endothelial cells, represents one of the earliest pathological changes in immune and inflammatory diseases such as atherosclerosis [72].

## 5. Conclusions

The use of nanoparticulate carriers in biomedicine is gaining popularity due to their ability to deliver drugs to specific biological targets, thereby addressing unmet medical and pharmaceutical needs. Our data demonstrate for the first time the important ability of ARC to significantly reduce endothelial cell activation and vWF release and suggest that its exploration in vasculopathies may be of particular interest.

## Figures and Tables

**Figure 1 pharmaceutics-14-00736-f001:**
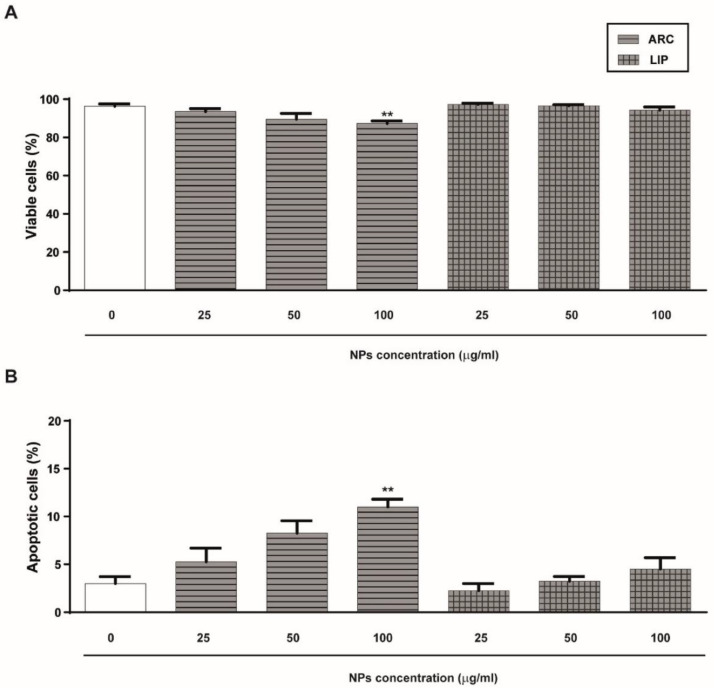
ARC do not alter HUVEC’s viability or proliferation. HUVEC were incubated for 24 h in the absence or presence of nanoarchaeosomes (ARC) or liposomes (LIP) at different concentrations. The percentage of viable cells (**A**) and apoptotic cells (**B**) was determined by staining cells with acridine orange and ethidium bromide (100 μg/mL) (one-way ANOVA followed by the Bonferroni multiple comparison tests; ** *p* < 0.01 vs. Basal. (**C**) The percentage of proliferating cells was determined after 24 h by measuring EdU incorporation in the viable subpopulation by flow cytometry. Results are the mean ± SEM of four to five independent experiments.

**Figure 2 pharmaceutics-14-00736-f002:**
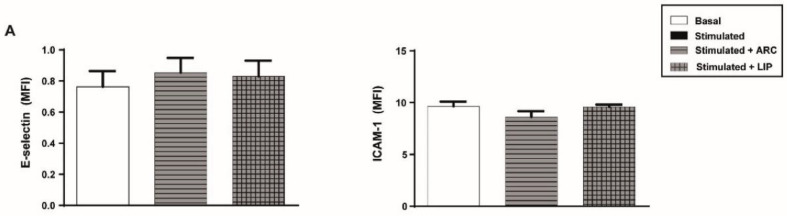
ARC selectively inhibit the expression of E-selectin and ICAM-1 on stimulated endothelial cells. HUVEC were treated or not with nanoarchaeosomes (ARC) or liposomes (LIP) (**A**). HUVEC were incubated with nanoarchaeosomes (ARC) or liposomes (LIP) at a concentration of 50 μg/mL and stimulated with (**B**) LPS (1 μg/mL); (**C**) Pam3CSK4 (1 μg/mL); (**D**) *E. coli* (MOI = 0.01); (**E**) TNF-α (0.3 μg/mL) or (**F**) IL-1β (0.1 ng/mL) Expression of E-selectin and ICAM-1 was evaluated 4 and 18 h after-stimulation, respectively (one-way ANOVA followed by Bonferroni multiple comparisons test); * *p* < 0.05, ** *p* < 0.01, *** *p* < 0.001 and **** *p* < 0.0001 vs. basal; # *p* < 0.05 and ## *p* < 0.01 between labeled groups. Results are the mean ± SEM of four to five independent experiments.

**Figure 3 pharmaceutics-14-00736-f003:**
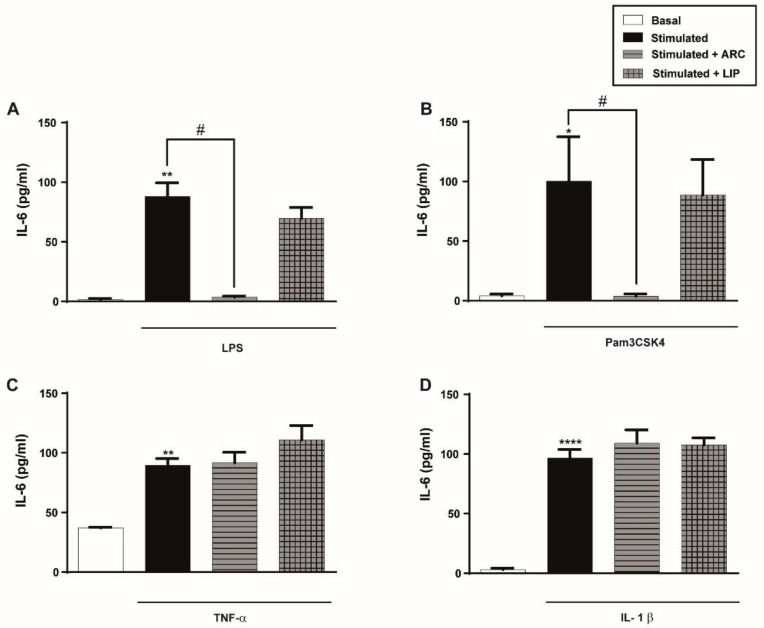
ARC selectively decrease the release of IL-6 by stimulated endothelial cells. HUVEC were stimulated with (**A**) LPS (1 μg/mL); (**B**) Pam3CSK4 (1 μg/mL); (**C**) TNF-α (3 ng/mL) or (**D**) IL-1β (0.1 ng/mL) and incubated with nanoarchaeosomes (ARC) or liposomes (LIP) at a concentration of 50 μg/mL for 18 h. Concentrations of IL-6 in supernatants were determined by ELISA (one-way ANOVA followed by a Bonferroni multiple comparisons test; * *p* < 0.05, ** *p* < 0.01 and **** *p* < 0.0001 versus basal; # *p* < 0.05 between labeled groups). Results are the mean ± SEM of four to five independent experiments.

**Figure 4 pharmaceutics-14-00736-f004:**
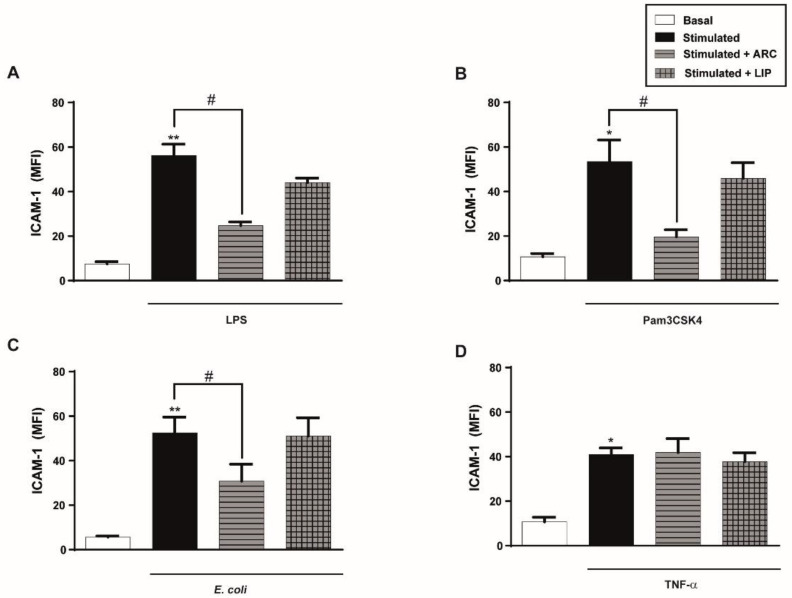
Preincubation with ARC also selectively inhibits ICAM-1 expression of endothelial cells. HUVEC were preincubated with nanovesicles (50 μg/mL) for 1 h, then washed and cells were stimulated with (**A**) LPS (1 μg/mL); (**B**) Pam3CSK4 (1 μg/mL); (**C**) *E. coli* (MOI = 0.01) or (**D**) TNF-α (0.3 ng/mL). ICAM-1 expression was evaluated 18 h poststimulation. (one-way ANOVA followed by Bonferroni multiple comparisons test; * *p* < 0.05 and ** *p* < 0.01 vs. basal; # *p* < 0.05 between labeled groups). Results are the mean ± SEM of 4 to 5 independent experiments.

**Figure 5 pharmaceutics-14-00736-f005:**
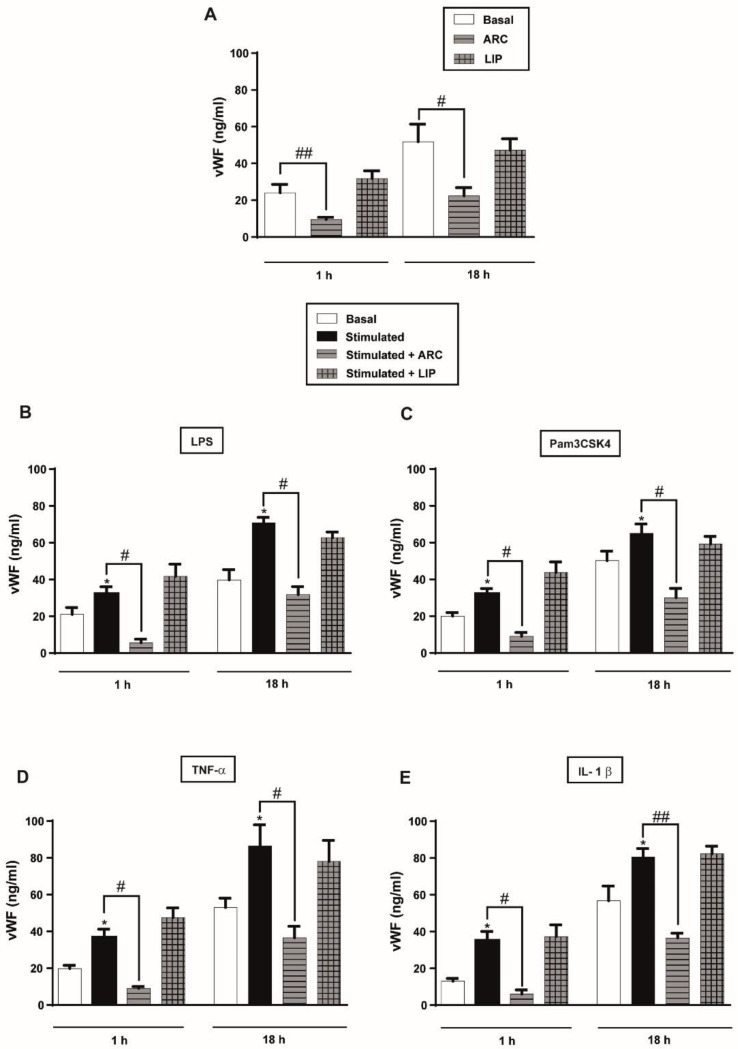
ARC decreases the amount of vWF released by endothelial cells. HUVEC were (**A**) unstimulated or stimulated with (**B**) LPS (1 μg/mL); (**C**) Pam3CSK4 (1 μg/mL); (**D**) TNF-α (3 ng/mL) or (**E**) IL-1β (1 ng/mL) and incubated with nanoarchaeosomes (ARC) or liposomes (LIP) at a concentration of 50 μg/mL for 1 and 18 h. The vWF concentrations in the supernatants were determined by ELISA (one-way ANOVA followed by Bonferroni multiple comparisons test; * *p* < 0.05 vs. basal; # *p* < 0.05 and ## *p* < 0.01 between the labeled groups). Results are the mean ± SEM of four to five independent experiments.

**Figure 6 pharmaceutics-14-00736-f006:**
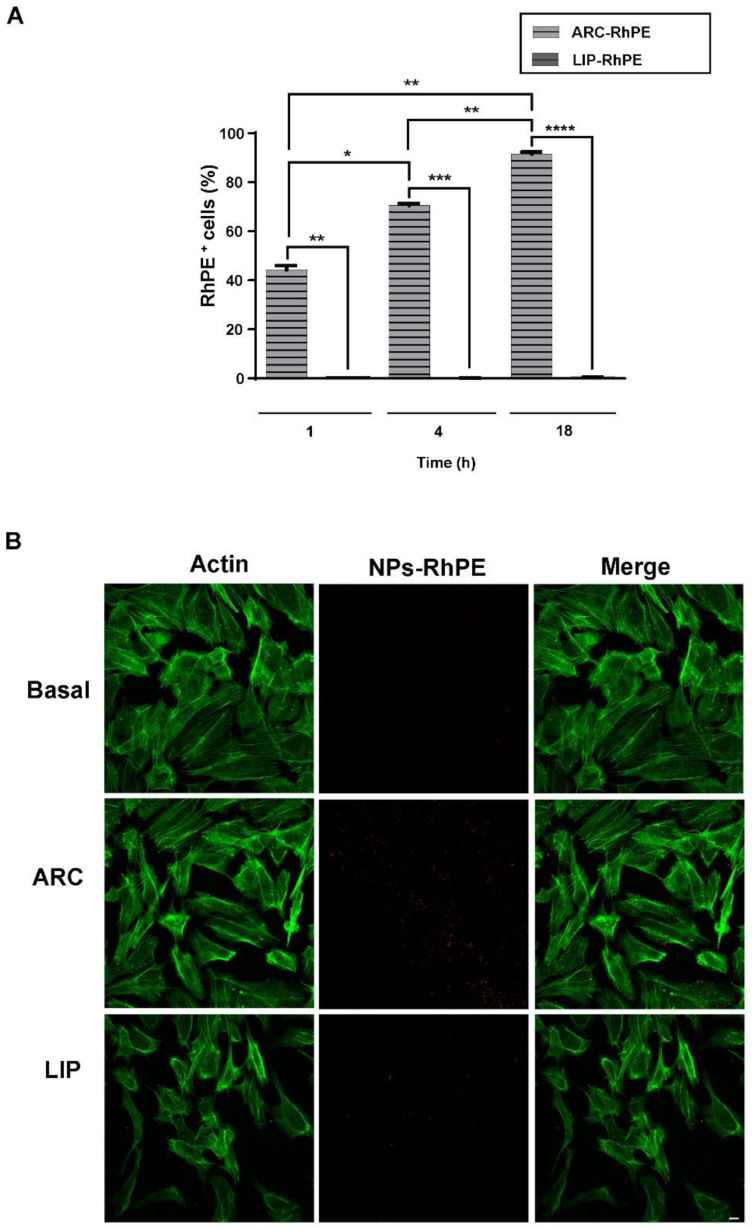
ARC are internalized by HUVEC. HUVEC were incubated for 1, 4 and 18 h with RhPE-labeled nanovesicles (ARC-RhPE and LIP-RhPE). Uptake assays were performed by (**A**) flow cytometry (one-way ANOVA followed by Bonferroni multiple comparisons test; * *p* < 0.05, ** *p* < 0.01, *** *p* < 0.01 and **** *p* < 0.001 between the labeled groups) and (**B**) Confocal microscopy. Representative images show RhPE-nanovesicles (red) and cytoplasm (green, FITC-phalloidin staining). Scale bars: 10 μm.

**Figure 7 pharmaceutics-14-00736-f007:**
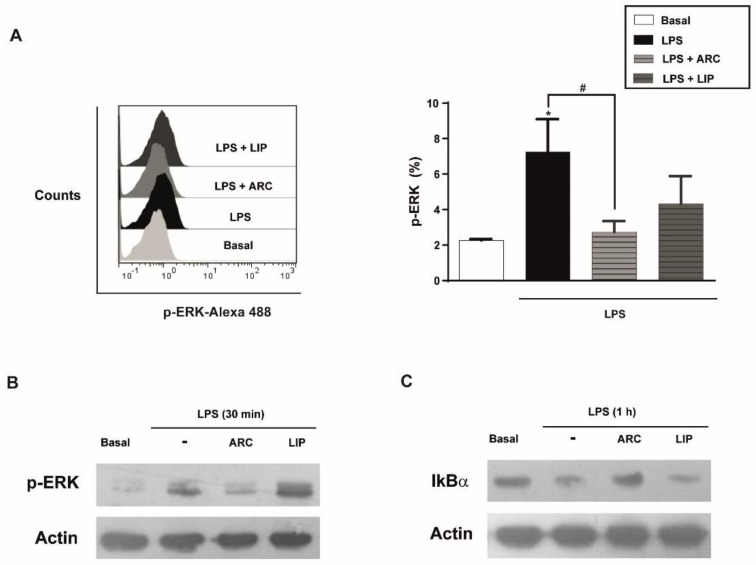
ARC attenuate LPS-induced phosphorylation of extracellular signal-regulated kinase (ERK1/2) and activation of NF-κB pathway in endothelial cells. (**A**) HMEC were stimulated with LPS (1 μg/mL) in the absence or presence of nanovesicles for 30 min, labeled with anti-p-ERK and analyzed by flow cytometry (one-way ANOVA followed by Bonferroni multiple comparisons test; * *p* < 0.05 vs. basal; # *p* < 0.05 between the labeled groups). HMEC were stimulated with LPS (1 μg/mL) in the absence or presence of nanoarchaeosomes (ARC) or liposomes (LIP) for 30 min (**B**) or 1 h (**C**) and cell extracts were analyzed by Western blot for p-ERK or IkBα, respectively. Representative images.

**Table 1 pharmaceutics-14-00736-t001:** Physicochemical characteristics of nanovesicles.

Formulation	Size (nm ± SD)	PDI	ζ Potential (mV ± SD)	TL (mg/mL ± SD)	RhPE (μg/mL ± SD)	RhPE/TL (μg/mg ± SD)
ARC	168.9 ± 11	0.19 ± 0.01	−41.5 ± 4.5	7.5 ± 1.3	-	-
LIP	276.7 ± 11	0.30 ± 0.08	−5.4 ± 1.4	7.0 ± 1.7	-	-
ARC-RhPE	159.7 ± 7.6	0.19 ± 0.02	−37.0 ± 4.0	6.9 ± 0.6	3.2 ± 0.6	0.48 ± 0.09
LIP-RhPE	251.1 ± 9.1	0.29 ± 0.07	−3.7 ± 0.6	6.0 ± 0.3	2.8 ± 0.9	0.45 ± 0.12

Data are expressed as mean ± standard deviation from five independent batches (*n* = 3 for RhPE labeled nanovesicles). PDI: Polydispersity Index; SD: Standard Deviation; TL: total lipids.

## Data Availability

Not applicable.

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
