# Peer review of "The Anti-Inflammatory Effect of Nanoarchaeosomes on Human Endothelial Cells"

_pharmaceutics, 2022, doi:10.3390/pharmaceutics14040736_

Round 1

Reviewer 1 Report

In the present study, authors investigated the anti-inflammatory effect of nanoarchaeosomes on human endothelial cells. The article has some questions as follows:

  1. In the abstract, “Halorubrum tebenquinchense” should be used in italics, and “E.coli” should be used in its full name for the first time, and used it in italics. Some words, such as “LPS”, “ICAM-1” should be used in its full name for the first time. And “NF-kB” should be “NF-κB”. In addition, each keyword should be separated by a semicolon.
  2. In Line 72, “Halorubrum tebenquinchense” should be used in its abbreviation “H. tebenquinchense”, the same as in the full manuscript.
  3. In the section of 2.3., there shouldn't be so many sections, the same as in the full manuscript.
  4. In Line 250-264, these sections should be deleted.
  5. In Table 1, a three-line form should be used. The contents in the table needed to be summarized and discussed in result 3.1.
  6. In Line 304, “E. coli” should be used in its full name for the first time, and used it in italics, and used in its abbreviation “E. coli” for the second time.
  7. In the abstract and conclusion, authors said that “Loaded with anti-inflammatory drugs, ARC could magnify their activity on inflamed endothelium, …”. However, authors didn’t load with any anti-inflammatory drugs, and didn’t investigate its effect combined with anti-inflammatory drugs. So, this description is incorrect in its current form.
  8. In Line 69-72, ARC's repair activity against inflammatory bowel and lung epithelium is due to coating/loading of the active substances dexamethasone and curcumin. Authors should be stated it with right form.
  9. In Fig. 6, the fluorescence image Merge of ARC and LIP does not show a significant difference. It is suggested to provide a clearer and complete field map or multiple fluorescence photography results as support.
  10. In section of 3.8., authors indicated that ARC could attenuate LPS-induced activation of NF-κB pathway. However, other proteins expression levels in NF-κB pathway were not showed, such as P65, P50, IKKα, and IKKα, p-P65, p-P50. The same as in the MAPK pathway, such as ERK, JNK, and P-38, and p-ERK, p-JNK, and p-P-38.
  11. The format of the reference is not correct.

Author Response

We are very grateful for the opportunity to resubmit a revised version of our manuscript entitled “The anti-inflammatory effect of nanoarchaeosomes on human endothelial cells” and for the positive responses of the reviewers and their constructive suggestions for improving the impact of the study. We have addressed all of the points raised, and are resubmitting an updated revised manuscript. We hope that the additions and changes meet with approval and that our study is now suitable for publication in Pharmaceutics.

1-In the abstract, “Halorubrum tebenquinchense” should be used in italics, and “E.coli” should be used in its full name for the first time, and used it in italics. Some words, such as “LPS”, “ICAM-1” should be used in its full name for the first time. And “NF-kB” should be “NF-κB”. In addition, each keyword should be separated by a semicolon.
All your suggested changes in the abstract have been done.

2-In Line 72, “Halorubrum tebenquinchense” should be used in its abbreviation “H. tebenquinchense”, the same as in the full manuscript.
H. tebenquinchense was used throughout all the ms after its first appearing in the text (not in the abstract) that is in line 73 of the revised ms.

3-In the section of 2.3., there shouldn't be so many sections, the same as in the full manuscript.
The number of sections has been edited throughout all the ms.

4-In Line 250-264, these sections should be deleted.
We apologize for this mistake; it has been deleted in the resubmitted ms.

5-In Table 1, a three-line form should be used. The contents in the table needed to be summarized and discussed in result 3.1.
The table has been changed to a three-line form and the data were summarized and discussed; lines 250-256. 

6-In Line 304, “E. coli” should be used in its full name for the first time, and used it in italics, and used in its abbreviation “E. coli” for the second time.
E.coli has been modified as suggested. In the text it is used for the first time in line 91.

7-In the abstract and conclusion, authors said that “Loaded with anti-inflammatory drugs, ARC could magnify their activity on inflamed endothelium, …”. However, authors didn’t load with any anti-inflammatory drugs, and didn’t investigate its effect combined with anti-inflammatory drugs. So, this description is incorrect in its current form.
We thank you for this comment. We agree that the vesicles were not loaded with anti-inflammatory drugs and our point was that because of the anti-inflammatory properties of the vesicles per se, the effect might be greater if they were loaded with an anti-inflammatory drug. We have reworded the sentences and hope it is now clearer. Abstract: lines 26-29 and discussion, lines;533-539.

8-In Line 69-72, ARC's repair activity against inflammatory bowel and lung epithelium is due to coating/loading of the active substances dexamethasone and curcumin. Authors should be stated it with right form.
Thank you for raising that point. The paragraph has been changed lines 73-74. 

9-In Fig. 6, the fluorescence image Merge of ARC and LIP does not show a significant difference. It is suggested to provide a clearer and complete field map or multiple fluorescence photography results as support.
Thank you very much for your comment. The images of Figure 6B were taken with 60x objective using a scanning zoom factor of 3x. We apologize for not making this clear in the legend of the figure. In these images, we have only focused on a couple of cells in order to have a better visualization of the nanoparticles. We appreciate your comment and therefore we decided to replace the original Figure 6B with a new Figure, which provides a complete field map that shows a significant difference in the merged image between ARC and LIP.

10-In section of 3.8., authors indicated that ARC could attenuate LPS-induced activation of NF-κB pathway. However, other proteins expression levels in NF-κB pathway were not showed, such as P65, P50, IKKα, and IKKα, p-P65, p-P50. The same as in the MAPK pathway, such as ERK, JNK, and P-38, and p-ERK, p-JNK, and p-P-38.
We thank the reviewer for these suggestions. We agree that there are many other signaling proteins of the NF-κB and MAPK pathways that we could have analyzed. However, we believe that the quantification of these few critical mediators is sufficient because this preliminary approach was not aimed to go deeper into the signaling pathway mechanism itself but to clearly highlight that the anti-inflammatory effect of ARC is related to the inhibition of the major signaling pathways involved in the activation of endothelial cells inflammatory responses.

11-The format of the reference is not correct.
We apologize for this error; The format’s references have been edited in the resubmitted version of the ms.

Reviewer 2 Report

Comments: major review

The authors have tried to investigate the anti-inflammatory effect of nanoarchaeosomes (ARC), on human endothelial cells. The authors reported that ARC (170 nm mean size, -41 mV ζ) did not affect the viability, cell proliferation, and expression of ICAM-1 and E-selectin under basal conditions but reduced the expression of both molecules and the secretion of IL-6 induced by LPS, Pam3CSK4 or E. coli. In addition, they reported that ARC significantly decreased basal von Willebrand factor (vWF) levels and those induced by all stimuli. The paper's subject is interesting, and the authors have tried performing several in vitro experiments to support their experimental findings. While this study falls within the scope of Pharmaceutics, there are several major concerns that must be addressed before further consideration:

  1. There is an unnecessary paragraph on pages 5 and 6, lines 250-264, which must be removed. This information is for the author only, no need to include it in the manuscript.
  2. On page 6, lines 267, 271, The structural features (size, polydispersity index, ζ potential, total lipids, and RhPE), better to replace the phrase structural features with “Physicochemical characteristics”
  3. The author claimed that, on page 3, lines 105-106, the reproducibility of each TPA-extract composition was routinely screened by phosphate content and ESI-MS, but there are no experimental results in the manuscript. The authors must provide those experimental results.
  4. Both cell viability and apoptotic results, page 7, at all tested concentrations show that ARC is relatively more toxic than the LIP, is it due to impurity or the ARC toxicity itself?
  5. Although the authors explained in the captions (Figure 2A as unstimulated), the Figure 2A legends must differ from Fig2B-2F.
  6. The authors tried to evaluated E-selectin and ICAM-1 expression at different times of post-stimulation, 4h, and 18h, respectively. Is there any reason to check at these two different times?
  7. The confocal microscopy image, Figure 6B, shows that unlike basal and LIP treated HUVEC, ARC has very noticeable effects on the HEVEC morphology and Actin fiber. This result is very interesting, and better if discussed more.

Author Response

We are very grateful for the opportunity to resubmit a revised version of our manuscript entitled “The anti-inflammatory effect of nanoarchaeosomes on human endothelial cells” and for the positive responses of the reviewers and their constructive suggestions for improving the impact of the study. We have addressed all of the points raised, and are resubmitting an updated revised manuscript. We hope that the additions and changes meet with approval and that our study is now suitable for publication in Pharmaceutics.

1-There is an unnecessary paragraph on pages 5 and 6, lines 250-264, which must be removed. This information is for the author only, no need to include it in the manuscript.

We apologize for this error; it has been deleted in the resubmitted version of the ms.

2-On page 6, lines 267, 271, The structural features (size, polydispersity index, ζ potential, total lipids, and RhPE), better to replace the phrase structural features with “Physicochemical characteristics”

The title of the table has been changed as suggested.

3-The author claimed that, on page 3, lines 105-106, the reproducibility of each TPA-extract composition was routinely screened by phosphate content and ESI-MS, but there are no experimental results in the manuscript. The authors must provide those experimental results.

The ESI-MS spectra have been included as supplementary Figure 1. 

4.Both cell viability and apoptotic results, page 7, at all tested concentrations show that ARC is relatively more toxic than the LIP, is it due to impurity or the ARC toxicity itself?

We thank the reviewer for this question. ARC are in fact relatively more cytotoxic than LIP. The reason is not the presence of impurities but a combination of the chemical nature of archaeolipids (sn 2,3 stereoisomerism, the mirror image of sn 1,2 phospholipids of LIP, ether rather than ester linkages to the glycerol backbone) and the extensive uptake of ARC by endothelial cells compared with the same LIP concentration.

5-Although the authors explained in the captions (Figure 2A as unstimulated), the Figure 2A legends must differ from Fig2B-2F.

Thank you for raising this point. We have reworded the legends in the resubmitted version of the ms, lines 312-317.

6-The authors tried to evaluated E-selectin and ICAM-1 expression at different times of post-stimulation, 4h, and 18h, respectively. Is there any reason to check at these two different times?

Thank you for your question. The different stimulation timing for E-selectin and ICAM-1 is due to the fact that in previous studies (Scholz D, Cell Tissue Res 1996; May MJ, Br J Pharmacol 1996 and Yamawaki M, J Dermatol Sci 1996), and in our own experience, the peak expression of E-selectin and ICAM-1 was 4 and 18-24 h post-TNF-stimulation of HUVEC respectively. Furthermore, E-selectin decreases to baseline levels after 24 h because protein-DNA interactions are lost at only one of the three NFkB-binding sites in the E-selectin promoter (Boyle EM, Journal of Surgical Research, 1999). This point was clarified in the M&M section, lines 182-187.

 7-The confocal microscopy image, Figure 6B, shows that unlike basal and LIP treated HUVEC, ARC has very noticeable effects on the HEVEC morphology and Actin fiber. This result is very interesting, and better if discussed more.

Thank you for drawing attention to this interesting topic. The images in the original Figure 6B were taken with a 60x objective and a scanning zoom factor of 3x. We apologize for not making this clear in the figure legend. In this condition, we had selected a group of cells from the ARC treatment that, we agree with you, appeared to have a random difference in the tension of their actin fibers. However, considering your comment, we took new whole-field confocal images with a magnification of 60x and found that there are no significant differences in the actin filament morphology under different conditions. Therefore, we have replaced Figure 6B with the new figure and we prefer to omit any comment on actin in the revised version of the ms.

Round 2

Reviewer 1 Report

It can be accepted in this current form.